# Colossal Magnetoresistance in Layered Diluted Magnetic Semiconductor Rb(Zn,Li,Mn)_4_As_3_ Single Crystals

**DOI:** 10.3390/nano14030263

**Published:** 2024-01-25

**Authors:** Yi Peng, Luchuan Shi, Guoqiang Zhao, Jun Zhang, Jianfa Zhao, Xiancheng Wang, Zheng Deng, Changqing Jin

**Affiliations:** 1Institute of Physics, Chinese Academy of Sciences, Beijing 100190, China; ypeng@iphy.ac.cn (Y.P.); g.q.zhao@iphy.ac.cn (G.Z.); zhaojf@iphy.ac.cn (J.Z.);; 2School of Physics, University of Chinese Academy of Sciences, Beijing 101408, China

**Keywords:** diluted magnetic semiconductor, colossal magnetoresistance, quasi-two-dimensional structure, single crystal

## Abstract

Diluted magnetic semiconductors (DMSs) with tunable ferromagnetism are among the most promising materials for fabricating spintronic devices. Some DMS systems have sizeable magnetoresistances that can further extend their applications. Here, we report a new DMS Rb(Zn_1−*x*−*y*_Li*_y_*Mn*_x_*)_4_As_3_ with a quasi-two-dimensional structure showing sizeable anisotropies in its ferromagnetism and transverse magnetoresistance (MR). With proper charge and spin doping, single crystals of the DMS display Curie temperatures up to 24 K. Analysis of the critical behavior via Arrott plots confirms the long-range ferromagnetic ordering in the Rb(Zn_1−*x*−*y*_Li*_y_*Mn*_x_*)_4_As_3_ single crystals. We observed remarkable intrinsic MR effects in the single crystals (i.e., a positive MR of 85% at 0.4 T and a colossal negative MR of −93% at 7 T).

## 1. Introduction

Diluted magnetic semiconductors (DMSs) have received extensive attention due to their significant potential for spintronic applications [1,2,3]. The intriguing properties of DMSs include their ability to mediate ferromagnetic interactions by tuning the conduction carriers and the strong coupling between the carriers and local spins [4,5]. The former can provide novel opportunities to control ferromagnetism, and the latter can lead to galvanomagnetic properties (e.g., magnetoresistance (MR) and anisotropic magnetoresistance effects) [6,7,8,9]. The most well-known examples of substantial magnetoresistance are magnetic multilayers and manganites, in which the coupling between the spin and charge is essential [10,11]. Similar coupling has generated anomalous negative MR in many ferromagnetic DMS materials (e.g., III-V based (Ga,Mn)As and II-II-V based (Ba,K)(ZnMn)_2_As_2_) [12,13]. However, most of the reported MR in DMS single crystals or single-phase thin films are relatively weak, with magnitudes of 10–30%.

The aforementioned (Ba,K)(ZnMn)_2_As_2_ has a layered structure to allow for the spatial and electronic separation of charge and spin doping [14]. It belongs to a new generation of DMSs in which the charges (or carriers) and spins are doped independently. The initial motivation to design and synthesize these new DMSs was to overcome the nonequilibrium Mn doping in the classical III-V based DMSs (e.g., (Ga,Mn)As, (In,Mn)Sb, etc.) [15]. This difficulty leads to limited solid solutions of Mn^2+^ in III-V-based DMSs. Consequently, the specimens of these DMSs are chemically metastable [16]. Ferromagnetic films are available only as thin films, and their material quality exhibits high sensitivity in preparation methods and annealing treatments. Eventually, the improvement in their Curie temperatures (*T*_C_) was prohibitive. Moreover, Mn substitution provided hole carriers together with local spins. Thus, one could not conduct electron doping to obtain *n*-type materials or *p*-*n* junctions for spintronic devices [17].

Taking advantage of the separated charge and spin doping, the (Ba,K)(ZnMn)_2_As_2_ achieved a reliable *T*_C_ of 230 K, which is close to room temperature [18]. Further studies stated that external compression on (Ba,K)(ZnMn)_2_As_2_ reduced the distance between the ZnAs layers and then decreased its *T*_C_ [19]. In other words, extending the distance of the ZnAs layers could result in a higher *T*_C_. RbZn_4_As_3_ has a well-defined quasi-two-dimensional layered structure with a larger ZnAs-layer distance (4.3 Å) than that of (Ba,K)(ZnMn)_2_As_2_ (3.7 Å). Although we did not obtain a high *T*_C_ after magnetic and charge doping, interesting galvanomagnetic properties were discovered. In this work, we report Rb(Zn_1−*x*−*y*_Li*_y_*Mn*_x_*)_4_As_3_ as a new type of DMS. We found remarkable MR effects on the single crystals, thus ruling out the possible influence of the grain boundaries in polycrystalline samples [20,21]. This DMS has a layered structure as the parent phase, which consists of infinitely stacked Rb layers and ZnAs layers. A detailed and comprehensive analysis of the critical behavior yielded the critical exponents *β*, *γ*, and *δ*. These exponents suggest that the mean-field model is the applicable theoretical model, indicating the long-range ferromagnetism in this material. Owing to the distinct anisotropic structure, sizeable anisotropies were observed in the ferromagnetism and transverse MR. Moreover, large positive MRs at low fields and colossal negative MRs at high fields exist in single crystals. It is worth noting that although large MRs were found in polycrystalline (Sr,K)(Zn,Mn)_2_As_2_ and (Ba,K)(Cd,Mn)_2_As_2_, one cannot rule out the contribution from the grain boundaries, which diffusely exist in the polycrystalline samples. These intriguing features of Rb(Zn_1−*x*−*y*_Li*_y_*Mn*_x_*)_4_As_3_ should benefit applications such as memory devices and magnetic sensors [3,22,23].

## 2. Materials and Methods

Polycrystalline specimens of Rb(Zn_1−*x*−*y*_Li*_y_*Mn*_x_*)_4_As_3_ were prepared via the solid-state reaction method with high-purity pristine reagents [17,24]. Firstly, Rb_3_As and Li_3_As were synthesized as precursors from Rb grains (99.9%), Li grains (99.9%), and As powder (99.99%). To avoid the oxidization and corrosion of the alkali metals (Rb and Li), raw materials were put directly into titanium tubes and sealed under an argon atmosphere at 1 bar. After that, the mixtures were heated slowly to 350 °C and then held for 5 h due to the low melting point and activity of the alkali metals. Afterwards, Zn powder (99.99%), Mn powder (99.99%), As powder, and the two precursors were mixed well and ground at the intended stoichiometric ratios. The mixtures were sealed in titanium tubes under an argon atmosphere after pelletization. The tubes were sealed in quartz ampoules before being sintered at 700 °C for 20 h. The collected materials were ground and pelletized before being sintered at 600 °C for another 20 h. This annealing process was necessary to complete the reaction and produce high-purity samples.

The growth of the single crystals was performed utilizing the self-flux method. High-purity elements, Rb grains, Zn powder, Li grains, Mn powder, and As powder were mixed at a ratio of 1:3.8−*y*:*y*:0.2:3. The mixture was then loaded in alumina crucibles, which were sealed in tantalum tubes and quartz ampoules. The ampoules were heated up to 1000 °C for 2 h and then quenched in ice water or liquid nitrogen after cooling the furnace temperature down to 930 °C. Note that all the processes were performed under the protection of high-purity argon. The entirety of a single crystal, after quenching, could be easily cleaved along its c-axis due to the quasi-two-dimensional layered structure of the titled material. Generally, the width of a perfect single crystal was only limited to the dimensions of the alumina crucible, and thus, the size of an available single crystal could be up to 10 mm × 7 mm × 1 mm.

Powder X-ray diffraction (XRD) was conducted with a Philips X’pert diffractometer (Malvern Panalytical Ltd, Malvern, UK) at room temperature to analyze the phase purity and structural parameters. Cu-K*α* radiation and 2*θ* scanning with a range from 10° to 120° were used during the XRD measurement. Energy dispersive X-ray analysis (EDX) with a commercial scanning electron microscope (SEM, Hitachi High-Tech Science Co., Ltd., Tokyo, Japan) was used to analyze chemical compositions of single crystals. Thus, the real atom ratios of single crystal samples are used in the following sections. In addition, considering the possible inhomogeneity of polycrystalline samples, we established the doping contents as the nominal ratio.

The characterization of *dc* magnetic susceptibility for all the samples was accomplished using a superconducting quantum interference device (SQUID, Quantum design, San Diego, CA, USA). The measuring temperatures were from 2 K to 300 K, and the measuring fields were up to 7 T. Quartz holders and brass holders with two quartz cylinders were used for the in- and out-of-*ab* plane (i.e., external field *H* parallel to the *ab*-plane and *c*-axis separately) magnetic measurement of single crystals, respectively. To ensure that the samples measured in the Arrott plot were initially magnetized, the isothermal magnetizations were measured after the samples were warmed up well above *T*_C_.

Electricity transport measurements were conducted with a physical property measurement system (PPMS, Quantum Design, San Diego, CA, USA). Similarly, the measuring temperatures were from 2 K to 300 K, and the measuring fields were up to 14 T. The single crystal samples were cleaved to obtain a clean, fresh surface for good ohmic contact. A standard four-wire method was employed to eliminate contact resistance with silver paint as an electrical contact and Pt wires as electrical leads. A current of 0.1 mA was used during all transport measurements. For Hall effect measurements, the layered Rb(Zn_1−*x*−*y*_Li*_y_*Mn*_x_*)_4_As_3_ single crystals were cut into thin flakes with a typical size of 5 mm × 1 mm × 0.05 mm. Below 10 K, Hall resistance was difficult to measure due to the oversized magnetoresistance from slightly asymmetric Hall contacts.

## 3. Results

### 3.1. Crystal Structure

The parent phase RbZn_4_As_3_ crystallizes into the quasi-two-dimensional tetragonal structure with the space group *P4*/*mmm* (No. 123, Z = 1), as shown in Figure 1a [25]. The structure is closely related to the *β*-BaZn_2_As_2_ or BaFe_2_As_2_ superconductor with a typical ThCr_2_Si_2_-type structure [26]. Similar to *β*-BaZn_2_As_2_, the lattice of RbZn_4_As_3_ consists of infinitely stacked Rb layers and ZnAs layers, which are based on edge-shared ZnAs_4_ tetrahedra. The double-stacked ZnAs_4_ tetrahedra along the *c*-axis distinguish RbZn_4_As_3_ from *β*-BaZn_2_As_2_.

Isovalent Zn^2+^/Mn^2+^ substitution was used to provide local spins, while Li^+^ was doped at the Zn^2+^ site for itinerate carriers. The nominal concentrations of Mn and Li can reach 20% in polycrystalline samples. Within this doping level, all of the peaks in the XRD patterns of polycrystalline samples with distinct Li and Mn doping levels can be indexed with a RbZn_4_As_3_ structure, suggesting that they share the same structure as the parent phase. To obtain the lattice constants *a* and *c*, Rietveld refinements were performed with GSAS software [27]. The refinement of the Rb(Zn_0.85_Li_0.1_Mn_0.05_)_4_As_3_ polycrystalline sample is plotted in Figure 1b as an example. According to a series of XRD patterns from varying Li- and Mn-doping polycrystalline samples, we successively obtained the cell volumes (*V*) of Rb(Zn_0.9−*x*_Li_0.1_Mn*_x_*)_4_As_3_, which are 180.85 (*x* = 0.05), 181.04 (*x* = 0.1), 181.05 (*x* = 0.15), and 181.51 (*x* = 0.2) Å^3^, while the *V* values of Rb(Zn_0.9−*y*_Li*_y_*Mn_0.1_)_4_As_3_ are 181.27 (*y* = 0.05), 181.04 (*y* = 0.1), 180.96 (*y* = 0.15), and 180.78 (*y* = 0.2) Å^3^. The results are plotted in Figure 1c, which demonstrates that the value of *V* monotonically increases with nominal Mn concentrations but decreases with nominal Li-doping levels, indicating successful chemical substitution. The XRD pattern of Rb(Zn_0.83_Li_0.1_Mn_0.07_)_4_As_3_ single crystals is shown in Figure 1d. Owing to the layered structure, all of the crystals are sheet-like and grow along the crystallographic *c*-axis. Thus, only the (0 0 l) peaks appear with 2*θ* scanning. As shown in the inset, the size of a typical sheet is around 2 × 1 × 0.02 mm^3^.

### 3.2. Magnetic Properties

For polycrystalline samples, the series of Rb(Zn_0.95−*y*_Li*_y_*Mn_0.05_)_4_As_3_ samples have the most significant magnetizations. Thus, we focus our discussion on these samples in the following sections. Figure 2a shows the temperature-dependent magnetization *M*(T) under zero-field-cooling (ZFC) and field-cooling (FC) processes of Rb(Zn_0.95−*y*_Li*_y_*Mn_0.05_)_4_As_3_ at an external field of 500 Oe. Rb(Zn_0.90_Li_0.05_Mn_0.05_)_4_As_3_ is nearly paramagnetic with a temperature down to 5 K. Correspondingly, the field-dependent magnetization *M*(H) of Rb(Zn_0.90_Li_0.05_Mn_0.05_)_4_As_3_ is a closed loop at 5 K. The slight S-shape of the loop indicates the presence of a short-range ferromagnetic correlation. More Li-doping induces stronger ferromagnetic-like behaviors. *M*(T) curves show *T*_C_ values of about 18 and 25 K for Rb(Zn_0.85_Li_0.10_Mn_0.05_)_4_As_3_ and Rb(Zn_0.80_Li_0.15_Mn_0.05_)_4_As_3_, respectively. With further increasing Li doping level to 0.2, the *T*_C_ slightly decreases to about 19 K. However, it is worth noting that the magnitude of magnetization of Rb(Zn_0.80_Li_0.2_Mn_0.05_)_4_As_3_ is much smaller than that of Rb(Zn_0.80_Li_0.15_Mn_0.05_)_4_As_3_. Furthermore, at 5 K, the hysteresis loop of Rb(Zn_0.80_Li_0.2_Mn_0.05_)_4_As_3_ becomes thinner than that of Rb(Zn_0.80_Li_0.15_Mn_0.05_)_4_As_3_, as shown in Figure 2b, indicating that an excessive doping level of Li damages the ferromagnetic ordering in the titled DMS materials. Notwithstanding its weak ferromagnetic interaction, it is significant for the contrast of the *M*(*T*) curve with a Li-free sample, which evidently displays paramagnetic behavior, as shown in Figure 2c. In short, the evolution of ferromagnetism with the Li doping level suggests that the itinerant carriers offered by the heterovalent substitution Zn^2+^/Li^+^ induce ferromagnetism.

The temperature-dependent inverse susceptibility (1/*χ*(*T*)) and corresponding Curie-Weiss fitting of Rb(Zn_0.85_Li_0.10_Mn_0.05_)_4_As_3_ are plotted in the inset of Figure 2a as a typical example. On the basis of high-temperature fitting, the effective magnetic moments (*M*_eff_) obtained via Curie-Weiss fitting (1/*χ* = (*T* − *θ*)/*C*) of the paramagnetic region are around 5 *μ*_B_/Mn [28]. The *M*(H) curves of the above three samples display open loops of ferromagnetism, as shown in Figure 2b. The coercive fields are about 0.2, 0.4, and 0.2 T, respectively.

Similar Li-doping-dependent behaviors can be found in single-crystal samples. Thus, the *M*(T) and *M*(H) curves of Rb(Zn_0.83_Li_0.10_Mn_0.07_)_4_As_3_ and Rb(Zn_0.78_Li_0.15_Mn_0.07_)_4_As_3_, which show robust ferromagnetism, are plotted in Figure 2d,e as typical examples. Distinct magnetic anisotropy can be found in both the *M*(T) and *M*(H) curves. It is clear that the easy axis of magnetization is along the *c*-axis. In Figure 2e, the saturation magnetizations (*M*_sat_) along the *c*-axis are 1.3 and 1.1 *μ*_B_/Mn for Rb(Zn_0.83_Li_0.10_Mn_0.07_)_4_As_3_ and Rb(Zn_0.78_Li_0.15_Mn_0.07_)_4_As_3_, respectively. On the other hand, the *M*_sat_ in the *ab*-plane is about 0.2 *μ*_B_/Mn. Both the *M*_sat_ and *M*_eff_ are comparable to those of (Ga,Mn)As and (Ba,K)(ZnMn)_2_As_2_ [29,30].

### 3.3. Magnetic Critical Behaviors

Thanks to homogeneous single crystals, we could obtain more accurate magnetization data in the vicinity of the *T*_C_ to analyze the critical behavior and calculate corresponding critical exponents. By analyzing the values of critical exponents, ferromagnetic interaction in Rb(Zn_1−*x*−*y*_Li*_y_*Mn*_x_*)_4_As_3_ can be determined. Firstly, the determination of *T*_C_ is indispensable and crucial. To obtain the precise *T*_C_, we analyzed the critical behaviors of two samples with the mean-field-behavior Arrott plot method as follows:*H*/*M* = *at* + *bM*^2^.(1)
where *a* and *b* are constants and *t* = (*T*_C_ − *T*)/*T*_C_ is the reduced temperature with an absolute value of |*t*| ≪ 1. In the Arrott plot, all the curves form a series of parallel lines in the high-field region. *T*_C_ can be determined when the intercept of the parallel line becomes zero [31,32,33]. The obtained *T*_C_ values are 20 and 24 K for Rb(Zn_0.83_Li_0.10_Mn_0.07_)_4_As_3_ and Rb(Zn_0.78_Li_0.15_Mn_0.07_)_4_As_3_ single crystals, respectively. Figure 3a–c are the Arrott plots of Rb(Zn_0.83_Li_0.10_Mn_0.07_)_4_As_3_ single crystals in the vicinity of the *T*_C_. It is worth noting that a short-range magnetic transition, like a spin-glass transition, cannot have a positive intercept in the Arrott plot. On the other hand, when we considered those modified Arrott plots in the situation of short-range exchange interaction, e.g., a three-dimensional Heisenberg model or a three-dimensional Ising model, the results markedly indicate a much lower *T_C_* (about 7 K) according to the generalized equation of state:(*H*/*M*)^1/^*^γ^* = *at* + *bM*^1/^*^β^*,(2)
where *β* and *γ* are critical exponents (*β* = 0.365 and *γ* = 1.386 for the 3D Heisenberg model; *β* = 0.325 and *γ* = 1.24 for the 3D Ising model; *β* = 0.5 and *γ* = 1 for the long-range mean-field model), as shown in Figure 3b,c, respectively, whereas the preceding mean-field model is much more consistent with the minimum of the derivative of magnetization d*M*(T)/d*T* in Figure 3d. For these three models, the high-field parts of the ideal Arrott plots are supposed to be parallel. In other words, they should show the temperature-independent slopes *K* in the high-field region. Thus, we defined the relative variation in slope Δ*K* = (*K*(T)/*K*(T_C_) − 1). Here, the slope *K*(*T*) is apparently temperature-dependent. Hence, the distribution of Δ*K* can effectively indicate which model is the most practical for samples to decide the critical exponents and determine the ferromagnetic interaction. Figure 3e shows the temperature dependence of the Δ*K* distribution. Obviously, the mean-field model possesses the most concentrated distribution at a value of 0, and the other two models have more dispersed distributions. This suggests that the mean-field model is the most suitable one.

Additionally, Kouvel–Fisher plots, another feasible and convenient analysis method to obtain relevant parameters in the vicinity of the critical temperature, were used to provide further evidence for our results [34,35]. On the basis of the critical equation of state, there are the following power law relations among magnetization *M*, susceptibility *χ*, reduced temperature *t* (or Curie temperature *T*_C_), and critical exponents:*M* ∝ *t^β^*, when *H* = 0 and *t* > 0;(3)
*χ* ∝ (−*t*)^−^*^γ^*, when *H*→0 and *t* < 0;(4)
*M* ∝*H*^1/*δ*^, when *H*→0 and *t* = 0, (5)
where *δ* = 1 + *γ*/*β* according to the critical scaling analysis. After applying a logarithm and differential to Equations (3) and (4), one can draw and fit the linear relations of −(dln*M*/d*T*)^−1^ versus *T* and −(dln*χ*/d*T*)^−1^ versus *T* in the critical region, i.e., Kouvel–Fisher plots, as
(dln*M*/d*T*)^−1^ = −(*T*_C_ − *T*)/*β*(6)
and
(dln*M*/d*T*)^−1^ = (*T*_C_ − *T*)/*γ*,(7)
in which the magnitude of the reciprocal of slope represents *β* and *γ*, respectively, and the horizontal intercept is *T*_C_. For the same sample, the Rb(Zn_0.73_Li_0.1_Mn_0.07_)_4_As_3_ single crystal, its Kouvel–Fisher plots are shown in Figure 3e,f. As shown in the two figures, in the narrow enough range of |*t*|, the fitting *β* and *γ* are 0.638 and 1.103, and the corresponding *T*_C_’s are 21.2 and 14.6 K, respectively. In order to guarantee that the fitting is carried out in the vicinity of the transition region, the selected data points for linear fitting must meet the requirement of |*t*| < 0.3. Despite the deviation of *T*_C_, the fitting results are closer to the former analysis based on the classic Arrott plot method. Meanwhile, it is feasible to determine another critical exponent, *δ,* by acquiring the *M*(*H*) data at *T*_C_ and plotting the corresponding log-log plot according to Equation (5). As shown in Figure 3h, the linear fitting gives *δ* = 2.705, in agreement with the value *δ* = 1 + *γ*/*β* = 2.729 from the obtained *β* and *γ*. Thus, our discussions about critical behavior are self-consistent, and the results confirm the long-range ferromagnetism of the single crystals.

### 3.4. Magnetoresistance and Hall Effect

The temperature dependence of resistivity (*ρ*(*T*)) of the aforementioned samples was measured. All of the samples exhibit semiconducting behavior, namely an increase in *ρ* with decreasing temperature. The conductivity is enhanced with increasing Li concentrations, which act as charge doping. As typical examples, Figure 2a shows the *ρ*(*T*) curves of ferromagnetic Rb(Zn_0.83_Li_0.10_Mn_0.07_)_4_As_3_ and Rb(Zn_0.78_Li_0.15_Mn_0.07_)_4_As_3_ single crystals. The decrease in resistivity is pronounced at low temperatures. Similar behaviors have been reported in other DMS materials, e.g., (Ga,Mn)As and Li(Zn,Mn)As [17,36]. ln*ρ* versus 1/*T* of Rb(Zn_0.83_Li_0.10_Mn_0.07_)_4_As_3_ single crystals is shown in Figure 4a. The band gap *E*_g_ ~19 meV is calculated from active model fitting:*ρ* = *ρ*_0_exp(*E*_g_/2*k*_B_*T*).(8)

This is only one-tenth of the parent compound (0.2 eV), which also indicates successful charge doping by Li substitutions.

Sizeable transverse magnetoresistance (fields perpendicular to current) can be found in ferromagnetic samples at low temperatures, ruling out the possible influence of grain boundaries in polycrystalline samples. The MR ratio is defined as MR ≡ (*ρ*(*H*) − *ρ*(0T))/*ρ*(0T) = Δ*ρ*/*ρ*(0T) [37]. Figure 4b shows this giant MR in the samples Rb(Zn_0.93−*y*_Li*_y_*Mn_0.07_)_4_As_3_, where *y* = 0.05, 0.1, 0.15, 0.2, among which the doping level of 10% Li contributes to the highest negative MR up to −93% at 7 T and the highest positive MR up to 85% at 0.4 T simultaneously. Given that MR changes slightly when the external field exceeds 7 T, the following measurements were only conducted within ±7 T. Figure 4c shows the temperature-dependent MR of Rb(Zn_0.83_Li_0.1_Mn_0.07_)_4_As_3_ single crystals along the *c*-axis and *ab*-plane. With the deceasing temperature, colossal MR appears in the vicinity of the Curie temperature *T*_C_ and enlarges monotonically. Meanwhile, the noticeable anisotropy of transverse MR is consistent with the magnetic anisotropy in Figure 2d,e. This indicates that the MR effect of ferromagnetic Rb(Zn_0.83_Li_0.10_Mn_0.07_)_4_As_3_ originates from a reduction in the carrier-spin scattering when local spins tend to be arranged completely parallel under external fields.

Figure 4d shows the field-dependent MR of Rb(Zn_0.83_Li_0.1_Mn_0.07_)_4_As_3_ single crystals at 2 K. With an external field along the *c*-axis, it displays a positive MR of 85% at 0.4 T and a negative MR of −93% at 7 T. It is worth noting that a positive MR with such a large magnitude has rarely been reported in previous DMS materials. In contrast, when using the *H*//*ab*-plane, the maximum values of the positive and negative MR become 7% and −35%, respectively. The saturation status of MR is consistent with the hysteresis loops. Magnetization and MR saturate at about 3 T along the easiest axis, namely the *c*-axis, while neither of them saturates even at 7 T. The positive MR can be attributed to the delayed rotation of spins under external fields, which corresponds to coercive fields (Figure 2e) [13]. Such a remarkable contrast of MR of low and high fields can stand for an off-and-on status and act as a magnetic switch in a circuit. Meanwhile, the large anisotropy of transverse MR makes the titled DMS a good candidate for magnetic field sensors [38].

To determine the carrier type and concentration, the Hall effect was measured. Figure 4e is the field-dependent Hall resistance (*ρ*_Hall_(*H*)) of Rb(Zn_0.83_Li_0.1_Mn_0.07_)_4_As_3_ single crystals below and above *T*_C_. Above *T*_C_, *ρ*_Hall_(*H*) curves are linear, and the carrier type can be easily determined as the *p*-type. Below *T*_C_, the *ρ*_Hall_(*H*) of a ferromagnetic conductor is expressed as
*R*_Hall_(H) = *R*_0_*H* + *R*_s_*M*,(9)
where *R*_0_ is the ordinary Hall coefficient, *R*_s_ denotes the anomalous Hall coefficient, and *M* denotes the magnetization of the samples [39,40]. At 10 K, the anomalous Hall component dominates the low-field region, so the carrier concentration is calculated with high-field data where *ρ*_Hall_(H) curves are linear. However, when the temperature falls below 10 K, the dramatically increasing MR will severely interfere with the Hall signals, and the anomalous Hall effect becomes unobservable. To avoid the influence of MR, Hall resistivity curves and the related calculations of carrier concentration are all based on the measurements of the Hall effect above 10 K. Similar to the DMS Li(Zn,Mn)As, *p*-type conduction is expected, as Li^+^ is doped to replaced Zn^2+^ to act as an acceptor. At 10 K, the carrier concentration is 3.5 × 10^19^ cm^−3^. It becomes 5.09 × 10^20^, 7.47 × 10^20^, and 7.48 × 10^20^ cm^−3^ at 50 K, 70 K, and 100 K, respectively. The temperature-dependent carrier concentration shows common features with semiconductors, i.e., a monotonic increase in the carrier concentration at low temperatures and saturation at a relatively high temperature, which is consistent with the typical thermal excitation model. The carrier concentration is comparable to that of (Ga,Mn)As or BaZn_2_As_2_ (~10^20^ cm^−3^). The exponential increase in hole density with increasing temperature is also consistent with the activation energy model behaviors of *ρ*(*T*) curves.

## 4. Conclusions

In summary, we synthesized a new diluted magnetic semiconductor Rb(Zn_0.83_Li_0.10_Mn_0.07_)_4_As_3_ with a quasi-two-dimensional structure. With optimal charge and spin doping, it shows ferromagnetic transition with the highest Cuire temperature of 24 K. The analysis of critical behavior via Arrott plots and reliable self-consistent results confirm the long-range ferromagnetic ordering in Rb(Zn_1−*x*−*y*_Li*_y_*Mn*_x_*)_4_As_3_ single crystals. Owing to the low-dimensional structure, substantial magnetic anisotropy can be found when the field is parallel and perpendicular to the *c*-axis. Correspondingly, transverse magnetoresistance also exhibits large anisotropy. The most remarkable feature is that the Rb(Zn_0.83_Li_0.10_Mn_0.07_)_4_As_3_ single crystals show a positive MR of 85% at 0.4 T and a negative MR of −93% at 7 T. It is worth noting that these intriguing features are found at low temperatures or high fields. These rigorous conditions could be barriers to practical applications. Nevertheless, Rb(Zn_1−*x*−*y*_Li*_y_*Mn*_x_*)_4_As_3_ could stimulate future developments of analog DMS materials, but with a near-room-temperature Curie temperature, to benefit applications.

## Figures and Tables

**Figure 1 nanomaterials-14-00263-f001:**
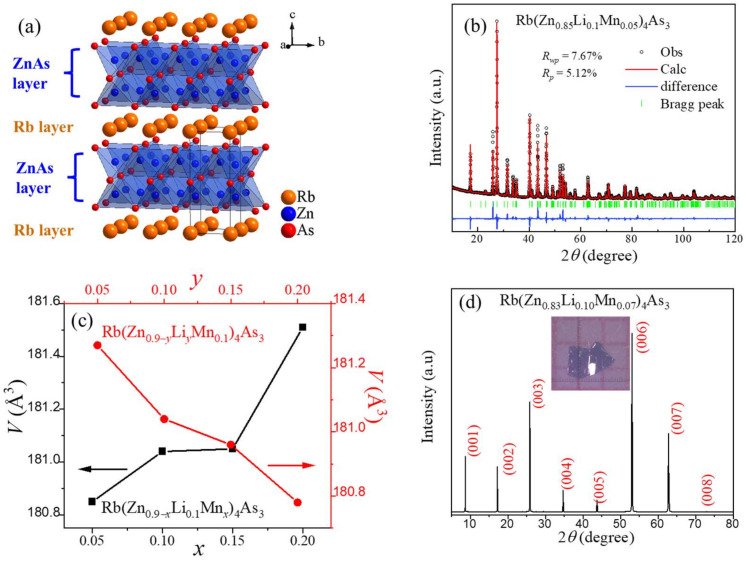
(**a**) Crystal structure of RbZn_4_As_3_ with stacking Rb layers and ZnAs layers. The black frame shows the unit cell. (**b**) The XRD pattern and the corresponding Rietveld refinement of Rb(Zn_0.85_Li_0.1_Mn_0.05_)_4_As_3_. (**c**) Lattice constants versus doping levels for Rb(Zn_0.9−*x*_Li_0.1_Mn*_x_*)_4_As_3_ and Rb(Zn_0.9−*y*_Li*_y_*Mn_0.1_)_4_As_3_, respectively. (**d**) The XRD pattern Rb(Zn_0.83_Li_0.1_Mn_0.07_)_4_As_3_ single crystal.

**Figure 2 nanomaterials-14-00263-f002:**
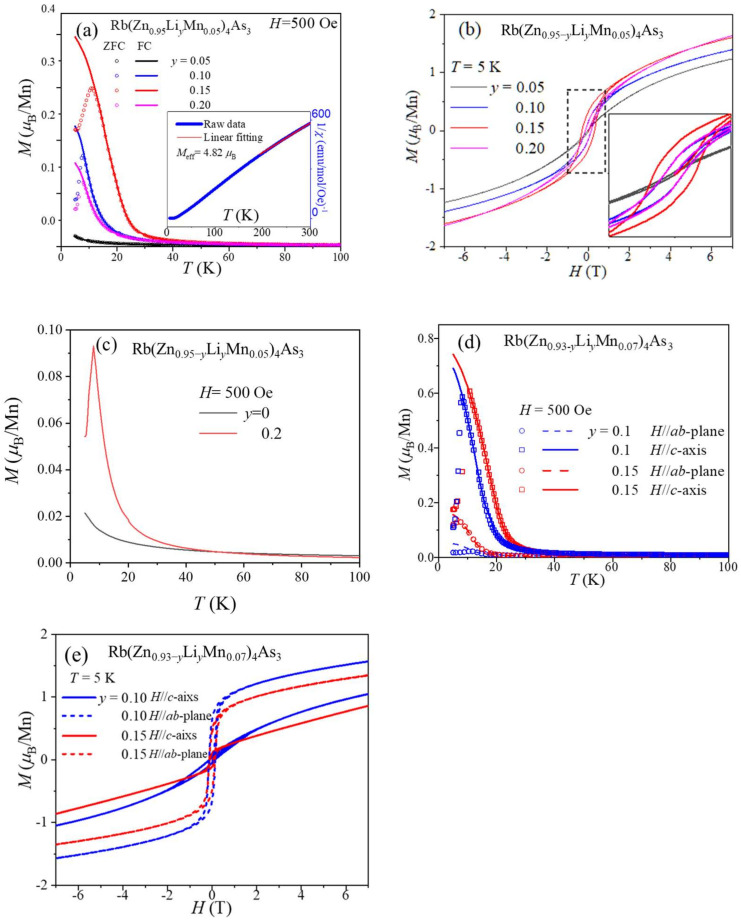
(**a**) Temperature-dependent magnetization of polycrystalline Rb(Zn_0.95−*y*_Li*_y_*Mn_0.05_)_4_As_3_ with *y* = 0.05, 0.10, 0.15, and 0.20 after ZFC and FC processes under an external field of 500 Oe. The inset is the temperature-dependent reciprocal of magnetic susceptibility and corresponding linear fitting. (**b**) Field-dependent magnetization of Rb(Zn_0.95−*y*_Li*_y_*Mn_0.05_)_4_As_3_ at 5 K. The inset is the enlarged magnetic loops at low fields. (**c**) *M*(T) curves of Rb(Zn_0.95_Mn_0.05_)_4_As_3_ and Rb(Zn_0.75_Li_0.10_Mn_0.05_)_4_As_3_ in the ZFC process. (**d**) *M*(T) curves of Rb(Zn_0.83_Li_0.10_Mn_0.07_)_4_As_3_ and Rb(Zn_0.78_Li_0.15_Mn_0.07_)_4_As_3_ single crystals with an external field parallel to the *c*-axis and *ab*-plane. (**e**) *M*(H) curves of Rb(Zn_0.83_Li_0.10_Mn_0.07_)_4_As_3_ and Rb(Zn_0.78_Li_0.15_Mn_0.07_)_4_As_3_ single crystals with an external field parallel to the *c*-axis and *ab*-plane at 5 K.

**Figure 3 nanomaterials-14-00263-f003:**
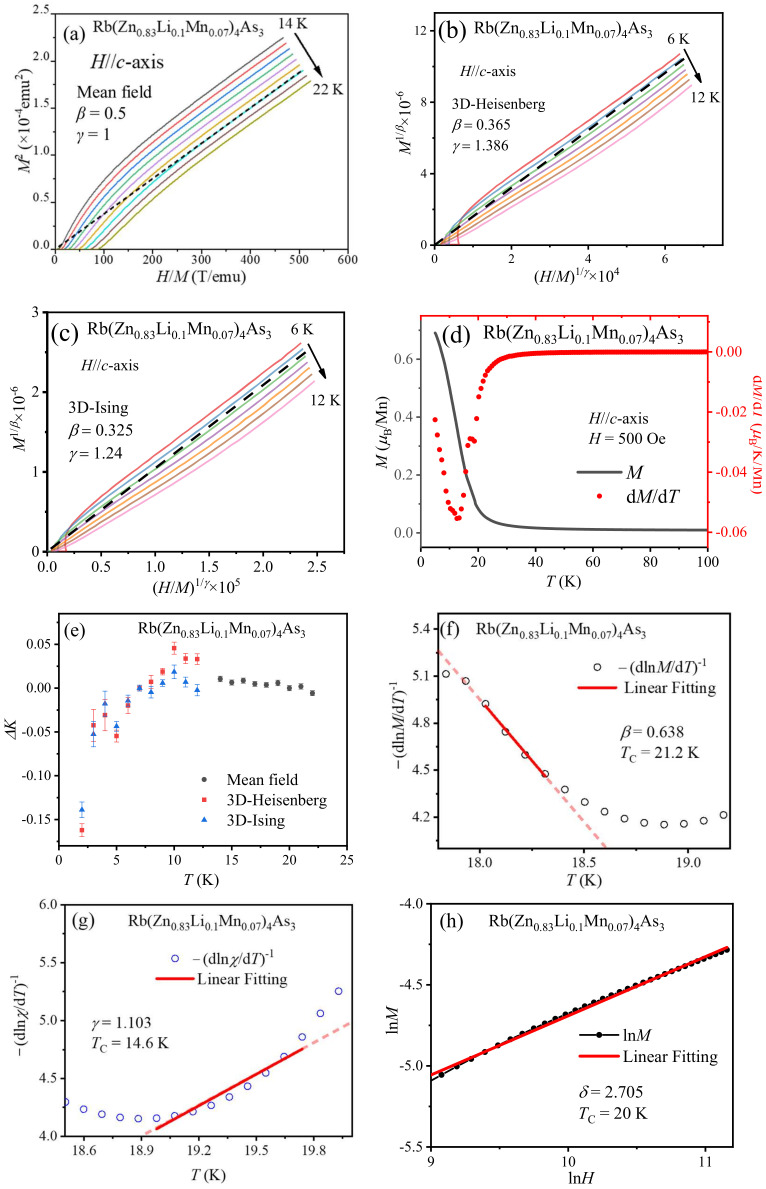
(**a**) Arrott plot of Rb(Zn_0.83_Li_0.10_Mn_0.07_)_4_As_3_ single crystals between 14 and 22 K with a step of 1 K for the mean-field model. (**b**) Arrott plot of Rb(Zn_0.83_Li_0.10_Mn_0.07_)_4_As_3_ single crystals between 6 and 12 K with a step of 1 K for the 3D Heisenberg model. (**c**) The 3D Ising model. (**d**) *M*(T) and d*M*(T)/d*T* curves of Rb(Zn_0.83_Li_0.10_Mn_0.07_)_4_As_3_ single crystals with an external field parallel to the *c*-axis. (**e**) The temperature-related distribution of the relative variation in slope from linear fitting equations of high-field Arrott plots. (**f**) Kouvel–Fisher plot of Rb(Zn_0.83_Li_0.10_Mn_0.07_)_4_As_3_ single crystals and corresponding linear fitting for *t* > 0. (**g**) Kouvel–Fisher plot of Rb(Zn_0.83_Li_0.10_Mn_0.07_)_4_As_3_ single crystals and corresponding linear fitting for *t* < 0. (**h**) Log-log plot at 20 K and corresponding linear fitting obtained from the *M*(H) plot at 20 K.

**Figure 4 nanomaterials-14-00263-f004:**
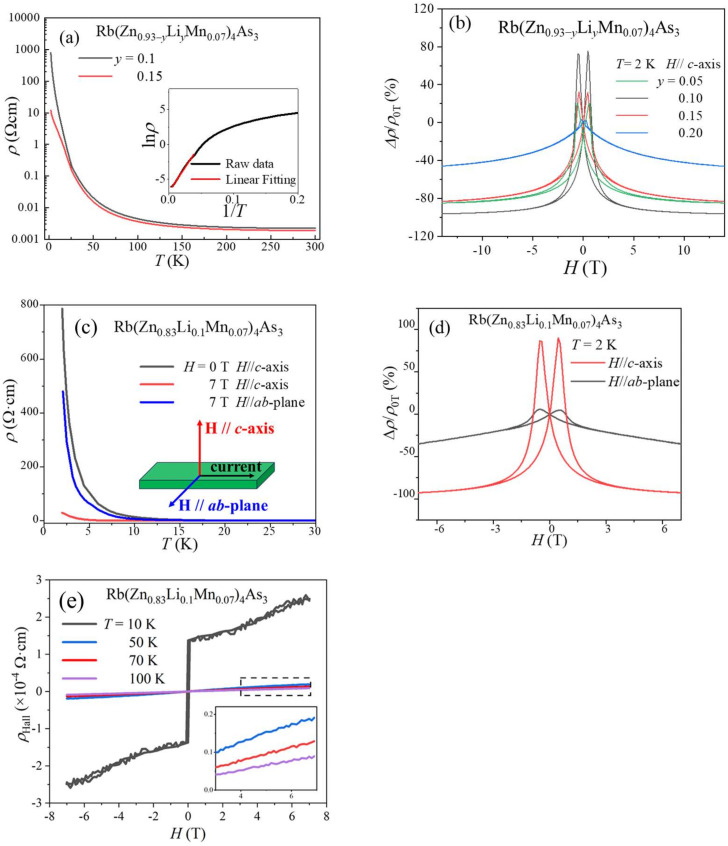
(**a**) Temperature-dependent resistivity of Rb(Zn_0.83_Li_0.10_Mn_0.07_)_4_As_3_ and Rb(Zn_0.78_Li_0.15_Mn_0.07_)_4_As_3_ single crystals. The inset is the active model fitting of Rb(Zn_0.83_Li_0.10_Mn_0.07_)_4_As_3_. (**b**) Transverse magnetoresistance of Rb(Zn_0.93−*y*_Li*_y_*Mn_0.07_)_4_As_3_ single crystals with H of 14 T parallel to the *c*-axis. (**c**) *ρ*(T) of Rb(Zn_0.83_Li_0.10_Mn_0.07_)_4_As_3_ single crystals with *H* of 7 T parallel to the *c*-axis and *ab*-plane at low temperatures. The inset is the configuration of the external fields and measured current. (**d**) Transverse magnetoresistance of the Rb(Zn_0.83_Li_0.10_Mn_0.07_)_4_As_3_ single crystals with *H* parallel to the *c*-axis and *ab*-plane at 2 K. (**e**) Hall effect measurement of Rb(Zn_0.83_Li_0.10_Mn_0.07_)_4_As_3_ single crystals. The inset is the enlarged high-field region.

## Data Availability

All data are available from the corresponding author upon reasonable request.

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
