# Peer review of "Colossal Magnetoresistance in Layered Diluted Magnetic Semiconductor Rb(Zn,Li,Mn)4As3 Single Crystals"

_nanomaterials, 2024, doi:10.3390/nano14030263_

Round 1
Reviewer 1 Report
Comments and Suggestions for Authors
The paper presents a complex work devoted to the development and characterization of diluted magnetic semiconductors (DMS) with quasi-two-dimensional structure that show ferromagnetic properties and large MR and Hall effect at low temperatures.
There are some comments.
1. The authors claim many times (Abstract, lines 199-202, 223-226) that such structures can be used as magnetic sensors, magnetic switches and have the potential to be used in future spintronic devices or even memory devices. However, the large MR and Hall effects are seen for temperatures lower than 20 K and for fields quite large and must be clearly emphasised that such structures are not working at room temperature. Also a quite large hysteretic behaviour can be seen. So, it is hard to believe that such practical application can be be implemented as they are already did with ferromagnetic thin films. So an elaborated comment concerning these issues and future developments of DMS in comparison with AMR, GMR and TMR structures must be made.
Lines 25-26: Magnetoelectric (ME) effect is characterized by appearance of an electric polarization (P) tempered by a magnetic field (H) or vice-versa - https://doi.org/10.1016/B978-0-12-803581-8.09207-9. In this case it is most appropriate to say "galvanomagnetic" effects like MR, AMR, and so on. Please correct.
Lines 90-91: How thick were the samples? cna you say a few words about the samples geometry/dimensions?
Line 162: "All the samples exhibit semiconducting conductivity" - please rephrase to avoid such words combination.
Line 163: Please use other expression like, for example, "The conductivity is enhanced... "
Line 169: it is wrong to say "active model". In fact it is about activation energy model which is expressed erroneously by equation 2
Equation 2: This equation is right but for conductivity! For resistivity the minus sign should be removed. Otherwise, resistivity will not decrease when T increases.
Lines 215-216 - please correct the term "active model"
A remark: for ferromagnetic samples, the state at zero field can be altered by the sample's history, i.e., previous magnetization states. Such that, more reliable is the saturation state, when all the magnetic moments are directed over the applied field. So, if the sample resistance is taken as a baseline in this state and, then, compared with the resistance for other fields (i.e. MR=(R(H)-R(Hsat)/R(Hsat) the MR effect will reach a maximum value at the coercive field. For some ferromagnetic samples, the MR ratio can be >0 or <0 in function of the relative direction between current and the applied field. So, this classification made by the authors that MR is >0 at small fields and <0 for large fields (i.e. the MR effect changes the sign) may be erroneous because they compare the resistance at zero field with R(H). So please argue your results.
Comments on the Quality of English LanguageThere are typo and grammar errors.
Some examples:
1. Several times the name Curie appears as Cuire (lines 41, 220)
2. Some phrases are difficult to be followed or understood: ex. lines 42-44, lines 88-90 ("Electricity transport" is a bad formulation, "Similarly, The...." So, please rephrase), lines 113-114 (please check for grammar - e.g., change ... increases.... decrease... by increases, decreases)
3. Line 183 - change as: "MR is defined as..." or using an appropriate formulation; Line 189 - probably the authors want to say "tend" not "trend"
Reviewer 2 Report
Comments and Suggestions for Authors
The authors report on a new diluted magnetic semiconductor (DMS) Rb(Zn1-x-yLiyMnx)4As3 with quasi-two-dimensional structure showing sizeable anisotropies in ferromagnetism and transverse magnetoresistance (MR). The results are interesting and original, the material is clearly presented, and conclusions are supported by the obtained results; however, the manuscript needs minor revision before publication.
Comments:
1) In the Introduction, the authors explain the advantage of separated charge- and spin-doping by giving example of (Ba,K)(ZnMn)2As2 in which a reliable TC of 230 K, which is close to room-temperature, was achieved. In the present study much lower transition temperatures were found. Therefore, the authors have to give more detailed motivation for synthesis of the new type DMS material and its chemical composition.
2) The authors consider magnetoresistance features and state, that “These intriguing features should benefit the applications as memory devices and magnetic sensors” (see the end of Introduction). This conclusion have to be made based on the obtained results, but the authors do not give any results on the application of the prepared samples for sensors or memory devices, so this statement has to be removed from the abstract (it could be left in the Introduction).
3) p.4: The authors claim that for polycrystalline samples, the series of Rb(Zn0.95-yLiyMn0.05)4As3 have most significant magnetizations and the transition temperatures increases with the increase of Li (y) up to 0.15, but again decrease for y=0.2. However, for single crystal samples the authors discuss the dependence on y content, but the results are presented for Mn x=0.07, not 0.05. Please explain in more details, why Mn content was different for the discussion in both cases. Moreover, it would be more informative for the reader to present also some data for polycrystalline films, not only write about the findings.
4) Please indicate the magnetic field values (not only “low” and “high” fields) when giving the MR values “single crystal shows positive MR of 85% at low fields and negative MR of -93% at high fields” (in the text and conclusions).
Comments on the Quality of English LanguageOnly minor editing of English language is required.
